# The Association between Maternal Periodontitis and Preterm Birth: A Case-Control Study in a Low-Resource Setting in Sudan, Africa

**DOI:** 10.3390/medicina58050632

**Published:** 2022-05-01

**Authors:** Lubna M. Shaggag, Nadiah ALhabardi, Ishag Adam

**Affiliations:** 1Dental Public Health Council, Sudan Medical Specialization Board, Khartoum 24984, Sudan; m.mugabil@mahdi.edu.sd; 2Department of Obstetrics and Gynecology, Unaizah College of Medicine and Medical Sciences, Qassim University, Unaizah 56219, Saudi Arabia; ia.ahmed@qu.edu.sa

**Keywords:** periodontitis, pregnancy, preterm birth, Sudan

## Abstract

*Background and Objectives:* Vast data have been published recently on the association between periodontitis and preterm birth (PB). However, these studies have shown inconsistent results. Few of them were conducted in Africa, and data has not been published on the association between periodontitis and PB in Sudan. *Materials and Methods:* A case-control study was conducted at the Omdurman maternity hospital in Sudan from February through October 2021. The cases were women with spontaneous PB (<37 weeks), and healthy women with TB (37–42 weeks) were the controls. Questionnaires (demographics, medical and obstetric factors) were completed through face-to-face interviews. Periodontitis was diagnosed by the Community Periodontal Index as: “bleeding on probing and a pocket depth of ≥3 mm and clinical attachment loss of ≥6 mm, calculus with plaque deposits, and gingival recession”. Multivariate regression analysis was performed with PB as the dependent variable. *Results:* One hundred sixty-five women were enrolled in each arm of the study. The age, parity and body mass index did not significantly differ between the women with PB and those with TB. Compared with the controls, a significantly higher number of women with PB had periodontitis (50/165 (30.3%) vs. 30/165 (18.2%), *p* = 0.011). The association between periodontitis and PB was significant. Women who had periodontitis had double the odds of having PB compared to women who had no periodontitis (adjusted Odd Ratio = 2.05, 95% Confidence Interval = 1.20–3.52). Moreover, the haemoglobin level (adjusted Odd Ratio = 0.67, 95% Confidence Interval = 0.51–0.88) was inversely associated with PB. *Conclusion:* The study results indicate that periodontitis and low haemoglobin were strongly associated with PB. Preventive measures, including the use of periodontitis screening and the prevention of anaemia, are needed to reduce PB in this setting.

## 1. Introduction

Preterm birth (PB) is defined as the birth of a baby before 37 completed weeks of gestation or 259 days from the final day of the last menstrual period [1]. It is a major worldwide health problem as there were 14.84 million PBs in 2014 [2]. Almost over three quarters (81.1%) of all PBs occur in South Asia and Africa [2]. PB is a major cause of perinatal mortality as well as a significant cause of long-term consequences among the survivors [3,4]. The prevalence of PB is high in sub-Saharan African countries; for example, 18.3% and 13.3% of births in Kenya and Ethiopia, respectively, were PBs [5,6]. Several factors such as age, [5,7], rural residency, lack of antenatal care [8], previous history of PB, multiple pregnancies and malaria infections [5] are reported to be associated with PB. The need to assess the risk factors of PB in different populations is high so that the World Health Organization (WHO) and United Nations’ 2010 goal of reducing mortality due to PB by 50% before 2025 can be achieved. The risk factors for PB should be correctly identified and properly managed to decrease the incidence of PB. Furthermore, identifying the risk of PB could help prevent PB if evidence-based preventive measures are implemented. Although the prevalence of PB is high in many African countries [5,6], pertinent information on the risk factors of PB are not adequately documented in Sub-Saharan Africa, including Sudan.

Vast data has recently been published on the association between periodontitis and PB [9,10,11,12,13,14,15,16,17,18]. However, these studies have shown inconsistent results, while some have shown that periodontitis was associated with PB [11,12,13,14]. Other studies have failed to show a significant association between periodontitis and PB [15,16,17,18]. Moreover, few of them were conducted in Africa [9,10], and no data has been published in Sudan on the association between periodontitis and PB. Perhaps the associations between periodontitis and PB in Sub-Saharan Africa, including Sudan, are influenced by other covariates such as anaemia, which is widely prevalent among pregnant women in Africa [19]. In Sudan, 24% of pregnant women had periodontal disease [20]. This study was conducted to investigate the association between periodontitis and PB in a tertiary hospital in Sudan.

## 2. Methods

A case-control study was conducted at the Omdurman maternity hospital in Sudan from February through October 2021. Omdurman is the largest governmental maternity hospital in Sudan; pregnant women from different socioeconomic levels attend daily and about 18,000 deliveries are performed per year. The hospital covers nearby localities and accepts referred patients from all over the country. The cases were women with spontaneous PB, and healthy women with TB (37–42 weeks) were the controls. Inclusion criteria (for both the cases and the controls) were mothers who delivered a single live baby. The exclusion criteria were post-term birth (≥42 weeks of gestation), unknown gestational age, multiple births, stillbirths, women with uterine anomalies and congenital malformed deliveries.

The systematic random sampling technique was followed to select the cases of PB and the controls (TB). The labour room records revealed 512 PBs in the nine months prior to the study. The sampling interval (≈3) was assumed by dividing the number of PBs (512) by the calculated sample size (512/165 ≈ 3). Thus, every three intervals of PB were taken until the required sample size (165) was reached. Normal women subsequent to the cases that had TB were taken as controls.

After signing an informed consent form, trained medical residents conducted face-to-face interviews with the mothers included in the study. Questionnaires on demographic, medical and obstetric factors were filled out in the local language (Arabic). The questionnaires recorded information concerning the mother’s age, parity, education, residence, occupation, antenatal care status, history of previous miscarriages/PBs, gestational age, haemoglobin level and the infant’s sex. Gestational age was calculated using both the dates of the last menstrual period and the early pregnancy ultrasound. Weight and height were measured following the standard procedure and were used to calculate the body mass index (BMI) as weight in kilograms divided by the squared height in metres. Following this, 2 mL of blood was withdrawn from every participant in an ethylenediaminetetraacetic acid and analysed for a complete blood count, including haemoglobin, using an automated haematology analyser and following the manufacturer’s instructions (Sysmex KX-21, Kobe, Japan). The results of the haemoglobin were recorded in the interview form. To prevent bias, the research team members were blind to the study status of the cases and controls.

The oral examination was performed by a single observer with previous experience in periodontics (main authors), utilising an odontoscope and a calibrated probe known as the “Community Periodontal Index” (CPI, with its basis being the three features of bleeding, dental calculus and gingival sulcus). Periodontal disease was diagnosed by recording gingival bleeding (BoP), the pockets’ (PD) all around dentition and attachment loss (LoA) around the index teeth (six in number) and the presence of calculus. Periodontitis was defined by the presence of a pocket depth greater than 3 mm on either the maxilla or the mandible or both, the presence of interdental clinical attachment loss (CAL) on the maxilla, the mandible or both of 2 mm or more and buccal or oral CAL of 3 mm or more [21]. The examiner was unaware of the mother’s gestational condition (PB/TD) at the time of the examination. 

### 2.1. Sample Size Calculation

A sample size of 165 women in each arm of the study was calculated for the case-control study (with ratio of 1:1) using the formula N = (r + 1/r) (($\dot{P}$) (1 − $\dot{P}$) (Z_β_ + Zα/2)^2^/(P1 − P2)^2^. N is the sample size in the case group, r is ratio of controls to cases and $\dot{P}$ is the measure of variability Z_β_, which represents the desired power. Zα/2 represents the desired level of statistical significance. P1 is the proportion in the case group. P2 is proportion in the control group. We have recently observed that 25% of pregnant Sudanese women had periodontitis [20]. Thus, we assumed that 40% of the cases and 25% of the controls had periodontitis (a difference of 15.0%). This sample was calculated to detect a difference of 5% at α = 0.05 with a power of 80%. The assumption was that 10% of the patients might not respond or might have incomplete data. The sample size was calculated using the OpenEpi Menu (Version 3.0 Sullivan, Atlanta, GA, USA) [22].

### 2.2. Statistics

The data were entered into a computer using the Statistical Package for the Social Sciences (SPSS) Statistics for Windows, version 22.0 (IBM, Armonk, NY, USA). Continuous data were checked for normality using the Shapiro–Wilk test and were found to be not normally distributed; these data were expressed as a median (interquartile [IQR]), while the categorised data were expressed as a frequency (proportion). Mann−Whitney U and Chi-square tests were used to compare non-parametric (continuous) data and proportions between women with PB and TB, respectively. Univariate analyses were performed with PB as the dependent variable and sociodemographic factors (age, residence and education), obstetric factors (parity, history of miscarriage/PB, antenatal care and BMI), haemoglobin level and peritonitis (present/absent) as the independent variables. Multicollinearity was evaluated by the presence of high correlations between the variables (r ≥ 0.9) or if the variance inflation factor was more than 4. There was no multicollinearity between the variables. Variables with their *p* < 0.200 were shifted to build the multivariable analysis and the backward likelihood ratio (LR) to evaluate the independent effects of each covariate by controlling the effects of other variables. The adjusted odds ratios (AOR) and 95% confidence intervals (CI) were computed. A *p*-value of less than 0.05 was considered statistically significant.

## 3. Results

One hundred sixty-five women were enrolled in each arm of the study. There was no significant difference in the median (IQR) age (28.0 [22.0–35.0] years vs. 30.0 [24.0–36.0] years, *p* = 0.1020, BMI (21.0 [18.8–22.8] kg/m^2^ vs. 21.3 [19.1–23.3] kg/m^2^, *p* = 0.305) of the women with PB and the women with TB. The median (IQR) of the haemoglobin level (11.3 [10.4–12.0] g/dL vs. 11.7 [11.2–12.3] g/dL, *p* < 0.001) was significantly lower in women with PB (Table 1).

There was no significant difference in the parity, residence, employment, antenatal care, history of miscarriage/PB, folic acid consumption in the first trimester, blood groups and newborn’s gender between the two groups. There was a significantly smaller number of educated women in the group of PB (Table 2). Compared with the controls, a significantly higher number of women with PB had periodontitis (50/165 (30.3%) vs. 30/165 (18.2%), *p* = 0.011) (Table 2).

Age, haemoglobin level, education, antenatal care and periodontitis were the factors whose *p* values were <0.020. Thus, these factors were shifted to develop the multivariate analysis. The results of the multivariate analysis showed that periodontitis (AOR = 2.05, 95% CI = 1.20–3.52) was associated with PB. Moreover, the haemoglobin level (AOR = 0.67, 95% CI = 0.51–0.88) was inversely associated with PB. There was no association between age, educational level, antenatal care and PB (Table 3).

## 4. Discussion

The main finding of the current study was that women with periodontitis were at 2.05 times higher risk for PB. Previous studies have shown that women who had periodontitis were at 6.3 and 2.3 times higher risk for PB in Uganda and Tanzania, respectively [9,10]. A recent (2021) review, which included 232 articles and 119,774 women, reported a statistically significant association between periodontal diseases and PB [11]. Recent reports utilizing large-scale claims data (748,792 pregnancy records) have shown that periodontitis was associated with an increased risk of PB (OR = 1.15) [12]. Moreover, in a large cohort which used Taiwan’s national medical records and included 1,757,774 pregnant women, the advanced and mild periodontal disease for PBs had an OR of 1.09 and 1.05, respectively [13]. A previous (2012) meta-analysis that included 22 studies enrolling 12,047 pregnant women showed that periodontitis was associated with increased OR for PB (the pooled OR odds ratio of PB −2.73 [95% CI: 2.06–3.6, *p* < 0.0001]) [14]. Twenty articles were included in the meta-analysis, which revealed a positive association between maternal periodontitis and PB (OR = 2.01) [23].

On the other hand, a case-control study, which included 148 cases and 296 controls, demonstrated no association between maternal periodontitis and PB [15]. In their recent (2020) systematic review of the evidence, Lavigne and Forrest failed to detect sufficient evidence for a causal relationship between periodontal disease and PB [24]. There was no association between periodontitis and PB in a small cohort study enrolling 39 pregnant women with periodontitis and 119 women without periodontitis [16]. Moreover, in a previous (2018) case-control study, maternal periodontal disease was not a risk factor associated with PB [17]. No significant association was previously reported between periodontal disease and PB [18].

The results of the current and former studies (above mentioned) should be compared with caution because of the differences between the studies regarding the sociodemographic characteristics, the prevalence of peritonitis, genetic differences and the methods used for diagnosing periodontitis. Moreover, the current study was a retrospective one, and this point is one of the limitations of our study.

PB was attributed to several pathological processes, mainly as an intra-amniotic infection. Periodontitis and its associated bacteria have been detected in the amniotic fluid, and these could be behind the spread of pathogens from the periodontal pockets [25]. It has been reported that an inflammatory process involving periodontal tissue, maternal serum and the vaginal compartment could explain the pathological mechanism involved in PB’s association with periodontitis [26]. Thus, in a recent meta-analysis that included 20 randomized controlled trials involving 8171 patients, periodontal treatment of infections (periodontal treatment) during pregnancy has been reported to reduce PBs [27]. However, a previous (2017) Cochrane database systematic review failed to detect a signficant asocation between periodontal treament and the reduction of PB risk [28].

In the current study, haemoglobin levels were inversely associated with PB (AOR = 0.67) and every g increase in haemoglobin level was associated with a 43.0% reduction in the risk of PB. We have previously reported that the risk of PB increases significantly with anaemia among pregnant women in eastern Sudan (OR = 3.2) [19]. In a recent (2020) meta-analysis of 58 studies including a total of 134,801 women, anaemic women were found to be at a higher risk of PB (AOR = 4.58) [7]. A previous (2019) meta-analysis of 117 studies including a total of 4,127,430 pregnancies showed that maternal anaemia was significantly associated with PB (OR = 2.11) [29]. In Kenya [6], anaemia was not found to be associated with PB.

## 5. Conclusions

The results indicate that periodontitis and low haemoglobin were strongly associated with PB. Preventive measures, including the use of periodontitis screening and the prevention of anaemia, are needed to reduce PB in this setting.

## Figures and Tables

**Table 1 medicina-58-00632-t001:** Univariate analysis of the factors (continuous variables) associated with preterm birth at Omdurman hospital in Sudan, 2021.

Variable	Preterm Birth (165)	Term Birth (165)	OR (95.0% CI)	*p*
Age, years	28.0 (22.0–35.0)	30.0 (24.0–36.0)	0.97 (0.94–1.01)	0.102
Body mass index, kg/m^2^	21.0 (18.8–22.8)	21.3 (19.1–23.3)	0.96 (0.88–1.04)	0.305
Haemoglobin level, g/dL	11.3 (10.4–12.0)	11.7 (11.2–12.3)	0.62 (0.48–0.80)	<0.001

OR = odds ratio. CI = confidence interval.

**Table 2 medicina-58-00632-t002:** Univariate analysis of the factors associated with preterm birth at Omdurman hospital in Sudan, 2021.

Variable		Preterm Birth (165)	Term Birth (165)	OR (95.0% CI)	*p*
		*Frequency (proportion)*			
Parity groups	Primipara	61 (37.0)	51 (30.9)	1.21 (0.74–1.97)	0.441
	Parous (2–5)	76 (46.1)	77 (46.7)	Reference	0.373
	Multipara (>5)	28 (17.0)	37 (22.4)	0.76 (0.42–1.37)	
Residence	Urban	63 (38.2.5)	54 (32.7)	Reference	0.321
	Rural	102 (61.8)	111 (67.3)	0.78 (0.50–1.23)	
Education level	≥secondary level	87 (52.7)	109 (66.1)	Reference	0.014
	<secondary level	78 (47.3)	56 (33.9)	1.47 (1.11–2. 72)	
Employment	Housewives	139 (84.2)	135 (81.8)	Reference	0.558
	Employees	26 (15.8)	30 (18.2)	0.84 (0.47–1.49)	
Blood group	O	82 (49.7)	74 (44.8)	Reference	0.378
	Other than O	83 (50.3)	91 (55.2)	0.82 (0.53–1.26)	
History of miscarriage/preterm birth	No	97 (58.8)	103 (62.4)	Reference	0.449
	Yes	68 (41.2)	62 (37.6)	1.16 (0.74–1.81)	
Taking iron/folic acid	Yes	24 (14.5)	20 (12.1)	Reference	0.518
	No	141 (85.5)	145 (87.9)	0.81 (0.42–1.53)	
Antenatal care	≥three visits	156 (94.5)	161 (97.6)	Reference	0.168
	<three visits	9 (5.5)	4 (2.4)	2.32 (0.70–7.69)	
Newborn gender	Male	88 (53.3)	87 (52.7)	Reference	0.912
	Female	77 (46.7)	78 (47.3)	0.97 (0.63–1.50)	
Periodontitis	No	115 (69.7)	135 (81.8)	Reference	0.011
	Yes	50 (30.3)	30 (18.2)	1.95 (1.16–3.27)	

OR = Odds ratio. CI = Confidence interval.

**Table 3 medicina-58-00632-t003:** Multivariate analysis of the adjusted factors associated with preterm birth at Omdurman hospital in Sudan, 2021.

Variable		AOR (95.0% CI)	*p*
Age, years		0.97 (0.94–1.01)	0.069
Haemoglobin level, gm/dL		0.67 (0.51–0.88)	0.004
Education level	≥secondary level	Reference	0.267
	<secondary level	1.31 (0.81–2.13)	
Antenatal care level	≥three visits	Reference	0.433
	<three visits	1.66 (0.46–5.89)	
Periodontitis	No	Reference	0.008
	Yes	2.05 (1.20–3.52)	

AOR = adjusted odds ratio. CI = confidence interval.

## Data Availability

Data are analyzed and included in this manuscript. The detailed data could be shared with the corresponded author upon a reasonable request.

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
