# Peer review of "The Association between Maternal Periodontitis and Preterm Birth: A Case-Control Study in a Low-Resource Setting in Sudan, Africa"

_medicina, 2022, doi:10.3390/medicina58050632_

Round 1

Reviewer 1 Report

Dear Authors,

I am pleased to read the assumptions, methodology and results of your research work.

The topic of your research has been discussed for over a dozen years, as evidenced by the data from your discussions and references. There are many findings in the current and historical literature regarding the relationship between preterm birth and maternal periodontitis. As I understand it, your study is to confirm these well-known results in the group of Sudanese patients and is of an epidemiological character confirming the already widely recognized dependence also in this narrow group. Apart from the remark about the low element of novelty, in your research I have also had other remarks about it.

  1. The title of the work should be clarified by adding the term “maternal” to periodontitis to avoid ambiguity
  2. In the abstract and the text of the work, you often use abbreviations without writing the full name beforehand
  3. I found the same sentence both in the abstract and in the text of the work: „ Eighty women had periodontitis; 50/165 (30.3%) vs. 30/165(18.2%), P=0.011, were in the groups of women who had periodontitis and in women who had no periodontitis, respectively.” It is unclear for me.Only a careful analysis of the Table 2 can tell the reader what you mean.I am asking you to rectify these ambiguities.
  4. In the hospital where you conducted the study, there are about 18,000 births, including over 10% (according to your data from the summary) preterm births, which means 1,800 preterm births per year. If you take into account the duration of the test on 9/12 years (from February till October 2021), i.e. 0.75 years, it should be at least 1350 preterm births in your hospital during the investigated time period. If you are more precise in the later part of materials and take into account the duration of 7 months as in the material and methods, it is still 1050 preterm births in your hospital during the investigated time period. You have half of this number, i.e. 512 PB in your research metods, why?
  5. How did you select the women to the control group for the study from the much larger number of patients in the studied period?
  6. Was the hemoglobin level determined, as you wrote, in the patient's interview?
  7. Wouldn't it be better to familiarize the reader, often not a periodontist, but a general practitioner or gynecologist, with the weight of individual periodontal indexes and their full names?
  8. In Sample size calculation, you wrote "Thus, we assumed that 40% of the cases and 25% of the control had periodontitis." I do not understand why? Before the test, did you have a result showing a 15% higher percentage of periodontitis in mothers with PB than in mothers with TB?
  9. In the results, however, you had 30% of PB with periodontitis and 18% of PB with periodontitis. Does this affect the sample size?
  10. Please show the complete sample size calculation methodology.
  11. Table 2 in my version has incompletely visible data, difficult to assess
  12. Why, in your conclusions. have you attributed a causal role to nutrition in terms of decreased hemoglobin levels? After all, there are many reasons for a decrease in hemoglobin levels. Have you found malnutrition to be the only factor in causing anemia in every case, you studied?

I do hope your answers will improve the scientific sound of your research before the publication.

Best regards

Author Response

We would like to thank the editor and the reviewers for their valuable comments on the manuscript. We feel that the comments have improved the manuscript dramatically.
Response to the reviewer # 1
Comment
 I am pleased to read the assumptions, methodology and results of your research work.

The topic of your research has been discussed for over a dozen years, as evidenced by the data from your discussions and references. There are many findings in the current and historical literature regarding the relationship between preterm birth and maternal periodontitis. As I understand it, your study is to confirm these well-known results in the group of Sudanese patients and is of an epidemiological character confirming the already widely recognized dependence also in this narrow group. Apart from the remark about the low element of novelty, in your research I have also had other remarks about it.
 Comment
1.    The title of the work should be clarified by adding the term “maternal” to periodontitis to avoid ambiguity
Response
The term maternal” has been added to periodontitis as suggested. Please see the title.
Comment
2.    In the abstract and the text of the work, you often use abbreviations without writing the full name beforehand
Response
The full name has been inserted as suggested. Please see line 12.
Comment

3.    I found the same sentence both in the abstract and in the text of the work: „ Eighty women had periodontitis; 50/165 (30.3%) vs. 30/165(18.2%), P=0.011, were in the groups of women who had periodontitis and in women who had no periodontitis, respectively.” It is unclear for me.Only a careful analysis of the Table 2 can tell the reader what you mean.I am asking you to rectify these ambiguities.
Response
Yes, it has been made in a clear manner. Please see line 20-21 and line 148-149.
Comment
4.    In the hospital where you conducted the study, there are about 18,000 births, including over 10% (according to your data from the summary) preterm births, which means 1,800 preterm births per year. If you take into account the duration of the test on 9/12 years (from February till October 2021), i.e. 0.75 years, it should be at least 1350 preterm births in your hospital during the investigated time period. If you are more precise in the later part of materials and take into account the duration of 7 months as in the material and methods, it is still 1050 preterm births in your hospital during the investigated time period. You have half of this number, i.e. 512 PB in your research metods, why?
Response
The details methods have been clarified as follow 
Generally, there are about 18,000 births in hospital. 
Then we checked the hospital records for PB in the previous nine months which were 512 Then we dived the expected number of PB (512) by the sample size (165) as (512/165 ≈ 3 . Thus , we took the cases with the interval of three. ie. One for every three. 
Please see line 68-72.
Comment
5.    How did you select the women to the control group for the study from the much larger number of patients in the studied period?
Response
Normal women subsequent to the cases and had TB were taken as control. Please see line 72
Comment

6.    Was the hemoglobin level determined, as you wrote, in the patient's interview?
Response
Yes it was measured and recorded in  the questionnaire. Please see line 86
Comment
7.    Wouldn't it be better to familiarize the reader, often not a periodontist, but a general practitioner or gynecologist, with the weight of individual periodontal indexes and their full names?
Response
The details have been inserted. Please see 90-97
Comment
8.    In Sample size calculation, you wrote "Thus, we assumed that 40% of the cases and 25% of the control had periodontitis." I do not understand why? Before the test, did you have a result showing a 15% higher percentage of periodontitis in mothers with PB than in mothers with TB?
Response
It was a summation, however the 25 % of the normal is our results in the previous work.
Comment
9.    In the results, however, you had 30% of PB with periodontitis and 18% of PB with periodontitis. Does this affect the sample size?
Response
We assumed the difference of 15% in the proportion of peritonitis between the cases and the control. However, our results showed difference of 12.0% and we do not think it has a major difference. 

Comment
10.    Please show the complete sample size calculation methodology.
Response
The complete sample size calculation has been shown. Please see line 100-110.
Comment
11.    Table 2 in my version has incompletely visible data, difficult to assess
Response
I think it is now visible. 
Comment
12.    Why, in your conclusions. have you attributed a causal role to nutrition in terms of decreased hemoglobin levels? After all, there are many reasons for a decrease in hemoglobin levels. Have you found malnutrition to be the only factor in causing anemia in every case, you studied?
Response
Yes, we agreed and it has been changed to anemia rather than malnutrition 

Reviewer 2 Report

  1. Consider revising the following: "Eighty women had periodontitis; 50/165 (30.3%) vs. 30/165(18.2%), P=0.011, were in the groups of women who had periodontitis and in women who had no periodontitis, respectively." as the groups are PB and TB.
  2. You mention that maternal age was not statistically different and then you add it in the multivariate analysis. Why?
  3. Please add a limitations paragraph in the manuscript concerning the retrospective study type.
  4. Otherwise, very interesting and well-written article.

Author Response

Response to the reviewer # 2
Comment
Consider revising the following: "Eighty women had periodontitis; 50/165 (30.3%) vs. 30/165(18.2%), P=0.011, were in the groups of women who had periodontitis and in women who had no periodontitis, respectively." as the groups are PB and TB.
Response
Yes, it has been made in a clear manner. Please see line 20-21 and line 148-149

Comment
You mention that maternal age was not statistically different and then you add it in the multivariate analysis. Why?
Response 
Although maternal age was not statistically different  but we planned in the method to  deal with as “variables with their P < 0.200 were shifted to build multivariable analysis and the backward likelihood ratio (LR) to evaluate the independent effects of each covariate by controlling the effects of other variables”. Please see line 124-126

Comment
Please add a limitations paragraph in the manuscript concerning the retrospective study type.
Response
It was inserted, please see line 193
Regards

This manuscript is a resubmission of an earlier submission. The following is a list of the peer review reports and author responses from that submission.